# Why Shape Coding? Asymptotic Analysis of the Entropy Rate for Digital Images

**DOI:** 10.3390/e25010048

**Published:** 2022-12-27

**Authors:** Gangtao Xin, Pingyi Fan, Khaled B. Letaief

**Affiliations:** 1Department of Electronic Engineering, Tsinghua University, Beijing 100084, China; 2Beijing National Research Center for Information Science and Technology, Tsinghua University, Beijing 100084, China; 3Department of Electrical and Computer Engineering, Hong Kong University of Science and Technology (HKUST), Hong Kong

**Keywords:** image compression, information theory, entropy rate, limit theorem, asymptotic bounds

## Abstract

This paper focuses on the ultimate limit theory of image compression. It proves that for an image source, there exists a coding method with shapes that can achieve the entropy rate under a certain condition where the shape-pixel ratio in the encoder/decoder is O(1/logt). Based on the new finding, an image coding framework with shapes is proposed and proved to be asymptotically optimal for stationary and ergodic processes. Moreover, the condition O(1/logt) of shape-pixel ratio in the encoder/decoder has been confirmed in the image database MNIST, which illustrates the soft compression with shape coding is a near-optimal scheme for lossless compression of images.

## 1. Introduction and Overview

One of Shannon’s outstanding achievements in source coding is to pointing out the ultimate data compression limit. This result has been widely and successfully applied in stream data compression. However, for image compression, it is still a challenging issue. This paper is an attempt to analyze the ultimate limit theory of image compression.

### 1.1. Preliminaries

Data compression is one of the basis of digital communications and helps to provide efficient and low-cost communication services. Images are the most important and popular medium in the current information age. Hence, image compression is naturally an indispensable part of data compression [1]. Moreover, its coding efficiency directly affects the objective quality of the communication network and the subjective experiences of users.

As a compression method with strict requirements, image lossless coding focuses on reducing the required number of bits to represent an image without losing any quality. It guarantees as large a reduction in the occupation of communication and storage resources as possible under certain system or scenario constraints. In the area of big data, image lossless coding may play a more significant role in applications in which errors are not allowed, such as in intelligent medical treatment, digital libraries, semantic communications [2,3], and metaverse in the future.

The entropy rate is an important metrics in information theory, which extends the meaning of entropy from a random variable to a random process. It also characterizes the generalized asymptotic equipartition property of a stochastic process. In this paper, we shall employ entropy rate to explain the best achievable data compression. It is well-known that the entropy rate of a stochastic process $\left\{Z_{i}\right\}$ is defined as
(1)$$H\left(Z\right)=\lim_{t\to \infty }\sup\frac{1}{t}H\left(Z_{1},Z_{2},\ldots ,Z_{t}\right).$$
If the limit exists, then H(Z) is the per symbol entropy of the *t* random variables, reflecting how the entropy of the sequence increases with *t*. Moreover, the entropy rate can also be defined as
(2)$$H^{\prime }\left(Z\right)=\lim_{t\to \infty }H\left(Z_{t}|Z_{t-1},Z_{t-2},\ldots ,Z_{1}\right).$$
$H^{\prime }\left(Z\right)$ is the conditional entropy of the last random variable given all previous random variables. For a stationary stochastic process, the limits in Equations (Equation 1) and (Equation 2) exist and are equal [4]. That is, H(Z) = $H^{\prime }\left(Z\right)$. In addition, for a stationary Markov chain, the entropy rate is
(3)$$\begin{array}{cc} H(Z) & =H^{\prime }\left(Z\right)=\lim_{t\to \infty }H\left(Z_{t}|Z_{t-1},\ldots ,Z_{1}\right) \end{array}$$
(4)$$\begin{array}{cc} \; & =\lim_{t\to \infty }H\left(Z_{t}|Z_{t-1}\right). \end{array}$$
The entropy rate is a long-term sequence metric. Even if the initial distribution of the Markov chain is not a stable distribution, it will still tend to converge as in Equations (Equation 3) and (Equation 4). Moreover, for a general ergodic source, the Shannon-McMillan-Breiman theorem points to its asymptotic equipartition property. If $\left\{Z_{i}\right\}$ is a finite-valued stationary ergodic process, then
(5)$$-\frac{1}{t}\log p\left(Z_{0},\ldots ,Z_{t-1}\right)\to H\left(Z\right)\;\mathrm{with}\;\mathrm{probability}\;1.$$
This indicates the convergence relationship between the joint probability density and entropy rate for the general ergodic process. Following a similar idea as that of the analysis of entropy rate, we investigate the asymptotic property of shape-based coding for stationary image ergodic processes.

### 1.2. Shape Coding

A digital image is composed of lots of pixels arranged in order. This form is fixed and if the size of an image is determined, the number and arrangement mode of the pixels is also determined. Shape coding extends the basic components of images from pixels to shapes, which is a more flexible coding method and may efficiently utilize image embedding structures. Additionally, it will no longer limit the number and position of shapes. Shape coding has three main characteristics: (1) The image is formed by filling shapes; (2) The position arrangement of shapes changes from a fixed mode to a random variable; (3) The shape database and codebook are generated in a data-driven way, which clearly contains more inherent features of image databases.

Consider a binary digital image *Z*, whose length and width are *M* and *N*, respectively, then the total number of pixels is t=M×N. Suppose this is divided into c(t) shapes $\left\{s_{1},s_{2},\ldots ,s_{c(t)}\right\}$, where $s_{i}$ is the *i*-th shape. We used D to denote the shape database and $F_{i}\left(S_{i}\right),i=1,\ldots ,T$ to represent filling an image with shape $S_{i}$ at position $\left(x_{i},y_{i}\right)$ in the *i*-th operation. The image with shape coding can be described as [5]
(6)$$\begin{array}{cc} \min & \sum_{i=1}^{c(t)}\left[l\left(s_{i}\right)+l_{p}\left(x_{i},y_{i}\right)\right] \end{array}$$
(7)$$\begin{array}{c} s.t.\;\;Z=\sum_{i=1}^{c(t)}F_{i}\left(s_{i}\right), \end{array}$$
where $l(s_{i})$ and $l_{p}\left(x_{i},y_{i}\right)$ represent the bit length of the shape $s_{i}$ and its corresponding location at $\left(x_{i},y_{i}\right)$, respectively. The constraint condition indicates that the binary image *Z* can be reconstructed through c(t) filling operations, which is exactly the same as the original image. On this premise, shape coding tries to reduce the cost required to represent an image as much as possible.

The codebook plays an important role in shape coding. It reflects the statistical characteristics and correlation of the data source. Figure 1 illustrates the structure of shape coding. It consists of two parts, namely the generation and use of the codebook. On the one hand, one searches and matches the shape of images in the dataset through a data-driven method. At the same time, the frequency statistical analysis is carried out to generate a shape database. The codebook can also be used repeatedly in communication and storage tasks to reduce the occupation of resources. The transmitter/compressor encodes the original image with the codebook. After transmission or storage through the channel/storage medium, the receiver/decompressor can decode the compressed file with the same codebook. In this way, one can completely reconstruct the original image in lossless mode.

### 1.3. Relations to Previous Work

The objective of this work is to present the performance limits from the perspective of information theory, which is related to our previous works in [5,6,7]. An image-encoding method through shapes and data-driven means can provide improvements in image lossless compression. In some known databases, *soft compression* outperforms the most popular methods, such as PNG, JPEG2000, and JPEG-LS. However, there is no theoretical support for how shape-based *soft compression* methods can reach the ultimate performance limit. That is, the gap between soft compression and its compression limit, namely the entropy rate is not theoretically known. However, the entropy rate associated with the asymptotic equipartition property analysis of images can help us design efficient encoding and decoding algorithms from the perspective of Shannon’s information theory.

The earliest multi-pixel joint coding method can be traced back to symbol-based coding [8], which transmits or stores only the first instance of each pattern class, and thereafter substitutes this exemplar for every subsequent occurrence of the symbols. This achieved a degree of bandwidth reduction on a scan-digitized printed text. Fractal theory [9,10] is also related to block-based coding. Fractal block coding approximates an original image by relying on the assumption that image redundancy can be efficiently exploited through self-transformability on a blockwise basis. However, *soft compression* generates the shape database in a data-driven manner, to create the codebook used in the encoder and decoder. Image processing-based data-driven methods such as [11,12,13] can explore the essential features of images and even eliminate semantic redundancy. The use of side information to assist data compression has also been used and analyzed by Kieffer [14] and Kontoyiannis [15]. Verdú [16] provided upper and lower bounds for the optimal guessing moments of a random variable by taking values on a finite set when the side information may be available. Rychtáriková et al. [17] generalized the point information gain and derived point information gain entropy, which may help analyze the entropy rate of an image.

Another relevent example is the Lempel-Ziv coding schemes [18]. These proposed the concept of compressibility. For every individual infinite sequence *x*, a quantity ρ(x) is defined. This is shown to be the asymptotically attainable lower bound on the compression ratio that can be achieved for *x* be any finite-state encoder. Wyner [19] derived theorems concerning the entropy of a stationary ergodic information source and used the results to obtain insight into the workings of the Lempel-Ziv data compression algorithm.

The main contribution of this paper is that we will be able to present a sufficient condition, which will allow for us to show that the performance limit of shape-based image coding can be asymptotically achievable in terms of entropy rate.

### 1.4. Paper Outline

The rest of this paper is organized as follows. Section 2 contains our main results, providing the asymptotic properties of shape-based image coding in terms of entropy rate. Moreover, we indicates the relationship between the numbers of shapes and coding performance. In Section 3, we present sample numerical results with concrete examples. In Section 4, we offer some complementary remarks and conclude this paper.

## 2. The Asymptotic Properties of Image Sources Composed of Shapes

The encoding method with shapes can take advantage of the characteristics of the data and simultaneously eliminate redundancy in the spatial and coding domains simultaneously. This section theoretically analyzes the performance of image coding with shapes. It will show that when the numbers of shapes and pixels have a reciprocal logarithm relationship, the average code length will asymptotically approach the entropy rate. To the best of our knowledge, this is the first result on image compression in information theory. The framework of this proof is similar to [4,19], but there are some important differences.

The average number of bits needed to represent the image *Z* with shapes are $B_{Z}$. Specifically,
(8)$$\begin{array}{cc} B_{Z} & =\frac{1}{t}\sum_{i=1}^{c(t)}\left[l\left(s_{i}\right)+l_{p}\left(x_{i},y_{i}\right)\right] \end{array}$$
(9)$$\begin{array}{cc} \; & \overset{\left(a\right)}{\leq }\frac{c\left(t\right)\log c\left(t\right)+\sum_{i=1}^{c(t)}l\left(s_{i}\right)}{t} \end{array}$$
(10)$$\begin{array}{cc} \; & \overset{\left(b\right)}{\leq }\frac{c(t)\log c(t)+c(t)\log|D|}{t} \end{array}$$
where (a) and (b) follow from the fact that the uniform distribution has maximum entropy. That is, $\sum_{i=1}^{c(t)}l\left(s_{i}\right)\leq c\left(t\right)\log D$ and $\sum_{i=1}^{c(t)}l_{p}\left(x_{i},y_{i}\right)\leq c\left(t\right)\log c\left(t\right)$. $B_{Z}$ is the average cost of encoding *Z*, which reflects the coding requirements of bits. In the sequel, we use Equation (Equation 10) instead of (Equation 8) to scale $B_{Z}$.

Let ${\left\{Z_{i}\right\}}_{i=-\infty }^{\infty }$ be a strictly stationary ergodic process with finite states and $z_{i}^{j}≜\left(z_{i},z_{i+1},\ldots ,z_{j}\right)$. Due to the invariance of time, $P(Z_{t}|Z_{t-k}^{t-1})$ is an ergodic process, where the *k*th-order Markov approximation is used to make an approximation. We will then have
(11)$$Q_{k}\left(z_{-(k-1)},\ldots ,z_{0},\ldots ,z_{t}\right)≜P\left(z_{-(k-1)}^{0}\right)\prod_{j=1}^{t}P\left(z_{j}|z_{j-k}^{j-1}\right),$$
where $z_{-(k-1)}^{0}$ is the initial state. In this way, one can use the *k*-th order Markov entropy rate to estimate the entropy rate of $\left\{Z_{i}\right\}$. That is,
(12)$$\begin{array}{cc} -\frac{1}{t}\log Q_{k}\left(Z_{1},Z_{2},\ldots ,Z_{t}|Z_{-(k-1)}^{0}\right)\; & =-\frac{1}{t}\log\prod_{j=t}^{t}P\left(Z_{j}|Z_{j-k}^{j-1}\right) \end{array}$$
(13)$$\begin{array}{cc} \; & =-\frac{1}{t}\sum_{j=1}^{t}\log P\left(Z_{j}|Z_{j-k}^{j-1}\right) \end{array}$$
(14)$$\begin{array}{cc} \; & \to -E\log P(Z_{j}|Z_{j-k}^{j-1}) \end{array}$$
(15)$$\begin{array}{cc} \; & =H(Z_{j}|Z_{j-k}^{j-1}). \end{array}$$
When k→∞, the entropy rate of the *k*th-order Markov approximation converges to the entropy rate of the original random process.

Suppose that $z_{1}^{t}$ is decomposed into c(t) shapes $s_{1},s_{2},\ldots ,s_{c(t)}$. We define $w_{i}$ as the *k* bits before $s_{i}$, where $w_{1}=z_{-(k-1)}^{0}$. Let $c_{lw}$ denote the number of shapes whose size is *l* and its previous state $w_{i}=w$, $w\in Z^{k}$.

**Lemma 1.** 
*For $\left\{Z_{i}\right\}$, the joint transition probability and shape size satisfy the following inequality*

(16)
$$\log Q_{k}\left(z_{1},z_{2},\ldots ,z_{t}|w_{1}\right)\leq \sum_{l,w}c_{lw}\log\frac{\alpha }{c_{lw}},$$

*where α is a constant.*


**Proof.** Suppose that for fixed *l* and *w*, the sum of the transition probabilities is less than a constant α, i.e.,
(17)$$\sum_{i:|s_{i}|=l,w_{i}=w}\frac{1}{c_{lw}}P\left(s_{i}|w_{i}\right)\leq \alpha .$$
Then,
(18)$$\begin{array}{cc} \log Q_{k}\left(z_{1},z_{2},\ldots ,z_{t}|w_{1}\right) & =\log Q_{k}\left(s_{1},s_{2},\ldots ,s_{c}|w_{1}\right) \end{array}$$
(19)$$\begin{array}{cc} \; & \overset{\left(a\right)}{=}\sum_{i=1}^{c}\log P\left(s_{i}|w_{i}\right) \end{array}$$
(20)$$\begin{array}{cc} \; & =\sum_{l,w}\sum_{i:|s_{i}|=l,w_{i}=w}\log P\left(s_{i}|w_{i}\right) \end{array}$$
(21)$$\begin{array}{cc} \; & =\sum_{l,w}c_{lw}\sum_{i:|s_{i}|=l,w_{i}=w}\frac{1}{c_{lw}}\log P\left(s_{i}|w_{i}\right) \end{array}$$
(22)$$\begin{array}{cc} \; & \overset{\left(b\right)}{\leq }\sum_{l,w}c_{lw}\log\sum_{i:|s_{i}|=l,w_{i}=w}\frac{1}{c_{lw}}P\left(s_{i}|w_{i}\right) \end{array}$$
(23)$$\begin{array}{cc} \; & \leq \sum_{l,w}c_{lw}\log\frac{\alpha }{c_{lw}} \end{array}$$
where (a) follows from Equation (Equation 11) and (b) follows from Jensen’s inequality, thanks to the convexity of logx for x>0. □

Lemma 1 links the conditional probability $Q_{k}\left(z_{1},z_{2},\ldots ,z_{t}|w_{1}\right)$ to $c_{lw}$, connecting the concepts before and after decomposing $\left\{Z_{i}\right\}$. We will continue to explore the quantitative relationship between shapes and pixels.

**Lemma 2.** 
*For $\left\{Z_{i}\right\}$, the number and size of its shapes meet the following relationship*

(24)
$$\sum_{l,w}c_{lw}\log\frac{c}{c_{lw}}\leq kc+\left(t+c\right)\log\left(1+\frac{c}{t}\right)+c\log\frac{t}{c}$$



**Proof.** For simplicity, we use *c* to represent c(t). Let $p_{lw}=\frac{c_{lw}}{c}$, then $\sum_{l,w}p_{lw}=1$. We define two random variables *U* and *V* such that
(25)$$\mathrm{Pr}\left(U=l,V=w\right)=p_{lw}.$$
The mean of *U* is the average length of each shape, i.e., $E\left(U\right)=\frac{t}{c}$. A random variable with a geometric distribution has maximum entropy when the mean of a discrete random variable is fixed. Thus, we have,
(26)$$\begin{array}{cc} H(U) & \overset{\left(a\right)}{\leq }\frac{t}{c}\log\frac{t}{c}-\left(\frac{t}{c}-1\right)\log\left(\frac{t}{c}-1\right) \end{array}$$
(27)$$\begin{array}{cc} \; & \overset{\left(b\right)}{\leq }\left(\frac{t}{c}+1\right)\log\left(\frac{t}{c}+1\right)-\frac{t}{c}\log\frac{t}{c} \end{array}$$
(28)$$\begin{array}{cc} \; & =\log\frac{t}{c}+\left(\frac{t}{c}+1\right)\log\left(\frac{c}{t}+1\right), \end{array}$$
where (a) is the entropy of a random variable with a geometric distribution and (b) follows that the function f(x)=xlogx−(x−1)log(x−1) is monotonically increasing when x≥1. On the other hand, $H\left(V\right)\leq {\log|Z|}^{k}=k$. Thus,
(29)$$\begin{array}{cc} \sum_{l,w}c_{lw}\log\frac{c}{c_{lw}} & =c\sum_{l,w}p_{lw}\log\frac{1}{p_{lw}} \end{array}$$
(30)$$\begin{array}{cc} \; & =cH(U,V) \end{array}$$
(31)$$\begin{array}{cc} \; & \leq cH(U)+cH(V) \end{array}$$
(32)$$\begin{array}{cc} \; & \leq c[\log\frac{t}{c}+\left(\frac{t}{c}+1\right)\log\left(\frac{c}{t}+1\right)]+kc \end{array}$$
(33)$$\begin{array}{cc} \; & =kc+\left(t+c\right)\log\left(1+\frac{c}{t}\right)+c\log\frac{t}{c}, \end{array}$$
which completes the proof. □

Based on these two lemmas, we will further analyze the condition under which the entropy rate can be reached asymptotically.

**Theorem 3.** 
*When the numbers of shapes and pixels meet the reciprocal relation of the logarithm, then the average encoding length will asymptotically approximate the entropy rate. That is,*

*If*

(34)
$$\frac{c(t)}{t}=O\left(\frac{1}{\log t}\right)$$

*then*

(35)
$$\lim_{t\to \infty }\frac{l(Z_{1},Z_{2},\ldots ,Z_{t})}{t}=H\left(Z\right).$$



**Proof.** From Lemma 1, one can write
(36)$$\begin{array}{cc} \log Q_{k}\left(z_{1},z_{2},\ldots ,z_{t}|w_{1}\right) & \leq \sum_{l,w}c_{lw}\log\frac{\alpha }{c_{lw}} \end{array}$$
(37)$$\begin{array}{cc} \; & =-\sum_{l,w}c_{lw}\log\frac{c\cdot c_{lw}}{c\cdot \alpha } \end{array}$$
(38)$$\begin{array}{cc} \; & =-c\log c-\sum_{l,w}c_{lw}\log\frac{c_{lw}}{c\alpha } \end{array}$$For simplicity, we use *Q* to represent $Q_{k}\left(z_{1},z_{2},\ldots ,z_{t}|w_{1}\right)$. Thus,
(39)$$\begin{array}{c} \frac{c}{t}\log c\leq -\frac{1}{t}\log Q-\frac{1}{t}\sum_{l,w}c_{lw}\log\frac{c_{lw}}{c\alpha } \end{array}$$From Lemma 2, it follows that
(40)$$\begin{array}{cc} -\frac{1}{t}\sum_{l,w}c_{lw}\log\frac{c_{lw}}{c\alpha } & =\frac{1}{t}\sum_{l,w}c_{lw}\log\frac{c}{c_{lw}}+\frac{c}{t}\log\alpha \end{array}$$
(41)$$\begin{array}{cc} \; & \leq \frac{1}{t}\left[kc+\left(t+c\right)\log\left(1+\frac{c}{t}\right)+c\log\frac{t}{c}\right]+\frac{c}{t}\log\alpha \end{array}$$
(42)$$\begin{array}{cc} \; & =\frac{c}{t}\left(k+\log\alpha \right)+\frac{c}{t}\log\frac{t}{c}+\left(1+\frac{c}{t}\right)\log\left(1+\frac{c}{t}\right). \end{array}$$When $\frac{c}{t}=O\left(\frac{1}{\log t}\right)$ and t→∞, the three terms in the right hand side of Equation (Equation 42) will all tend towards to 0. Combining Equations (Equation 39) and (Equation 42), we obtain
(43)$$-\frac{1}{t}\sum_{l,w}c_{lw}\log\frac{c_{lw}}{c\alpha }\to 0,\;\;\mathrm{when}\;t\to \infty .$$Then,
(44)$$\begin{array}{cc} \lim_{t\to \infty }\sup\frac{c(t)\log c}{t} & \leq \lim_{t\to \infty }-\frac{1}{t}\log Q_{k}\left(Z_{1},Z_{2},\ldots ,Z_{t}\right|Z_{-(k-1)}^{0}) \end{array}$$
(45)$$\begin{array}{cc} \; & \to H(Z). \end{array}$$The asymptotic property of the second term in the right hand side of Equation (Equation 10),
(46)$$\begin{array}{c} \lim_{t\to \infty }\frac{c(t)\log|D|}{t}=0. \end{array}$$Thus,
(47)$$\begin{array}{cc} \lim_{t\to \infty }\frac{l(Z_{1},Z_{2},\ldots ,Z_{t})}{t} & =\lim_{t\to \infty }\left(\frac{c(t)\log c}{t}+\frac{c(t)\log|D|}{t}\right) \end{array}$$
(48)$$\begin{array}{cc} \; & =\lim_{t\to \infty }\frac{c(t)\log c}{t} \end{array}$$
(49)$$\begin{array}{cc} \; & =H(Z). \end{array}$$This shows that when c(t) and *t* meet the condition in Equation (Equation 34), the average coding length of $\left\{Z_{i}\right\}$ will asymptotically approximate the entropy rate H(Z). □

Theorem 3 sets up a bridge between the shapes and the entropy rate for image sources with ergodic properties. This theoretically indicates what order of magnitude we should use to obtain the shapes and pixels. When one encodes images with shapes, the average cost will asymptotically tend toward the entropy rate if the numbers of shapes and pixels satisfy the reciprocal relation of the logarithm. Moreover, this provides new insights into the design of image compression algorithms in theory.

## 3. Numerical Analysis

Section 2 points out the asymptotic property of encoding methods based on shapes. When $\frac{c(t)}{t}\to O\left(\frac{1}{\log t}\right)$, the average encoding length will asymptotically approximate the entropy rate. This indicates the relationship between the shape-pixel number ratio and coding performance. In this section, we present some numerical results to illustrate that for each ergodic process of an image source, if $\frac{c(t)}{t}\to O\left(\frac{1}{\log t}\right)$ as t→∞, one can obtain the result of Equation (Equation 35).

Table 1 reveals the numerical results on the MNIST datasets. This includes encoding results $R_{avg}$ and $\frac{1}{t}c\left(t\right)\log t$ in ten categories with the soft compression algorithm [5]. What can be clearly seen in this table is that $\frac{1}{t}c\left(t\right)\log t<1$ for all classes. This is on the order of O(1), which is consistent with the assumption in Theorem 3.

We focused on simulated images as an alternative analysis. We used the birth and death processes of two states to simulate a stationary ergodic process. For each case, 5000 $\left\{Z_{i}\right\}$ with M=100,N=100 were generated, respectively. We encoded $\left\{Z_{i}\right\}$ with fixed size shapes and observed the effect of $\frac{c(t)}{t}$ on coding performance.

Figure 2 illustrates the shape coding working mechanism of the image source. This indicates the performance of the encoding method with shapes, in bits per pixel (bpp). Cases 1–5 represent different parameters of the infinitesimal generator matrix of the birth-death process, illustrating the relationship between coding performance and $\frac{c(t)}{t}$. In different cases, the change trend of these curves is the same. The bpp decreases with the increase in shape size (i.e., the shape-pixel number ratio decreases), which reflects the gain brought by shape. Moreover, as the shape-pixel number ratio continues to decrease, bpp enters the smoothing region. This also shows that the reduction in the number ratio will not always improve the encoding performance. This is due to the fineness of the model itself, which does not take advantage of the additional statistical information of larger shapes. Note that, the numerical difference between the curves is essentially the difference of the entropy rate.

## 4. Concluding Remarks

In this paper, we investigated the performance limit of shape-based image compression. Our works answered the open problem regarding the relationship between image decomposition and lossless compression, which reflects the performance variation in general. Specifically, when the numbers of shapes and pixels have a reciprocal relation to the logarithm relation, the average code length will asymptotically approach the entropy rate.

For image coding algorithms, one should pay full attention to the superiority of shapes in image processing. Likewise, it is necessary to take advantage of the characteristics of the image dataset. Through shapes and data-driven means, one can use the high-dimensional information of images to help with coding. Moreover, the asymptotic analysis of the entropy rate can also be extended to gray images and multi-component images, with some adjustments.

Finally, it is noted that this paper focuses on the source part, without considering the natural robustness of images in the communication process. In a future work, we will explore the theory of joint source-channel image coding in the finite block length regime. It is noted that image lossless compression, especially, soft compression, may become an important block for semantic information communications, and even play some roles in the new developments of metaverse-type services in the future.

## Figures and Tables

**Figure 1 entropy-25-00048-f001:**
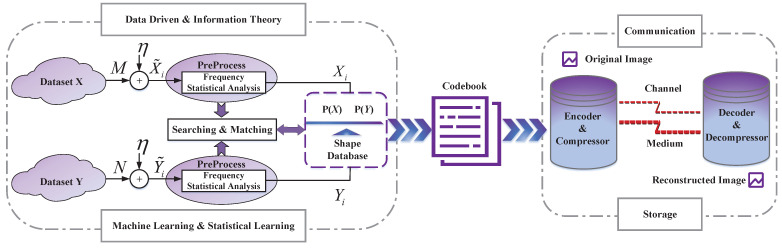
The structure of shape coding. It consists of two parts, namely the generation and use of the codebook. The former makes use of the characteristics of the data source, while the latter improves the compression efficiency.

**Figure 2 entropy-25-00048-f002:**
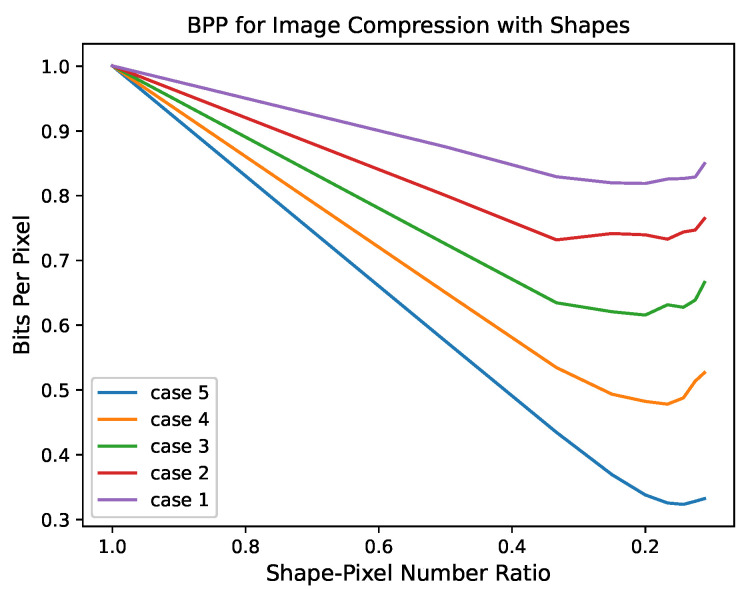
The performance of encoding method with shapes, in bits per pixel (bpp).

**Table 1 entropy-25-00048-t001:** The numerical analysis of shapes and pixels on MNIST dataset ($R_{avg}$ is the average compression ratio).

Class	0	1	2	3	4	5	6	7	8	9
$R_{avg}$	2.84	6.02	3.17	3.20	3.77	3.40	3.20	4.05	2.81	3.52
$\frac{1}{t}c(t)\log t$	**0.200**	**0.080**	**0.178**	**0.175**	**0.149**	**0.163**	**0.175**	**0.136**	**0.202**	**0.157**

## Data Availability

Not applicable.

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
