# Peer review of "Why Shape Coding? Asymptotic Analysis of the Entropy Rate for Digital Images"

_entropy, 2022, doi:10.3390/e25010048_

Round 1

Reviewer 1 Report

The article presents an interesting approach to the problem of image coding.

The main problem of the article is terminological. The practical approach which they use to demonstrate their findings is not an image but handwritten letters deposited in the database MNIST.  The application of the method on real compressed images, which the authors widely discuss in the introduction, would probably bring quite differernt results. The usage of real images for testing is for me condition for publication of this article.

Generally the article seems to be mathematically consistent.

The entropy of an image based on assumption of pre-existent shapes with an multifractality assumption was previously reported by Rychtáriková, R.; Korbel, J.; Macháček, P.; Císař, P.; Urban, J.; Štys, D. Point Information Gain and Multidimensional Data Analysis. Entropy 2016, 18, 372. https://doi.org/10.3390/e18100372, . Authors should discuss this article. I am sure that if the authors seriously analyse real images, they would come to the same conclusion, i.e. that Shannon entropy is insufficient to characterise the image and a spectrum of parametric entropy is needed for this task.

Reviewer 2 Report

In this paper, the authors provide a theoretical analysis of the soft compression method that they had previously proposed. In general, the paper provides some insight into soft compression and tries to provide a mathematical proof for the main theorem on the asymptotical approximation of the average encoding length. I have carefully reviewed the proof. I am most concerned, however, about the correctness of the proof that leads to the main conclusion: "for an image source, there exists a coding method with shapes that can achieve the entropy rate under new findings, a certain condition where the shape-pixel ratio in the encoder/decoder is O(1/logt)"  . In addition, several other comments also need to be addressed before the work can be considered further.

Major comments:

1. In equation 4 of page 6, it is stated that "the three terms in the right hand side of equation (34) will all tend to 0", but the second term may not necessarily tend to 0. As a result, equation 35-41 cannot be obtained. As a consequence, the main theorem can be flawed.

Minor comments:

1. The only simulation result presented in the paper does not provide sufficient information. For cases 1-5, the authors needed to add specific conditions in order to better understand the trends.

2. If there is no such term as "ultimate limit theory," consider replacing it with theoretical analysis or another appropriate term.

3.  There are a number of symbols that lack well-defined descriptions, which will impact the paper's readability. Please review the manuscript carefully and add definitions where necessary. For example, Fi in equation 7, alpha in equation 16, and Q in equation 32 are not defined.

3. It is necessary to proofread the paper. There are a few grammatical errors here and there, such as line 45-46, it should be "through data-driven method" rather than "through data-driven".

Round 2

Reviewer 1 Report

Thank you for revision of the manuscript.